

# Comparison of the toxic effects of different mycotoxins on porcine and mouse oocyte meiosis

Yujie Lu[1], Yue Zhang[1], Jia-Qian Liu[1], Peng Zou[1], Lu Jia[1], Yong-Teng Su[2], Yu-Rong Sun[2] and Shao-Chen Sun[1]

[1] College of Animal Science & Technology, Nanjing Agricultural University, Nanjing, China
[2] Jiangsu Aomai Bio-Tech Company, Nanjing, China

## ABSTRACT

**Background:** Aflatoxin B1 (AFB1), deoxynivalenol (DON), HT-2, ochratoxin A (OTA), zearalenone (ZEA) are the most common mycotoxins that are found in corn-based animal feed which have multiple toxic effects on animals and humans. Previous studies reported that these mycotoxins impaired mammalian oocyte quality. However, the effective concentrations of mycotoxins to animal oocytes were different.

**Methods:** In this study we aimed to compare the sensitivity of mouse and porcine oocytes to AFB1, DON, HT-2, OTA, and ZEA for mycotoxin research. We adopted the polar body extrusion rate of mouse and porcine oocyte as the standard for the effects of mycotoxins on oocyte maturation.

**Results and Discussion:** Our results showed that 10 μM AFB1 and 1 μM DON significantly affected porcine oocyte maturation compared with 50 μM AFB1 and 2 μM DON on mouse oocytes. However, 10 nM HT-2 significantly affected mouse oocyte maturation compared with 50 nM HT-2 on porcine oocytes. Moreover, 5 μM OTA and 10 μM ZEA significantly affected porcine oocyte maturation compared with 300 μM OTA and 50 μM ZEA on mouse oocytes. In summary, our results showed that porcine oocytes were more sensitive to AFB1, DON, OTA, and ZEA than mouse oocytes except HT-2 toxin.

## INTRODUCTION

Mycotoxins are secondary metabolites produced by fungi, while the most agriculturally common mycotoxins known today include aflatoxins (AF), deoxynivalenol (DON), ochratoxin A (OTA), zearalenone (ZEA) (*Grajewski et al., 2012*). These mycotoxins have multiple toxic effects on human and animal health in a very low dose, which draws worldwide attention.

Aflatoxins are the mycotoxins which widely exist in corn-based animal feed. Considering the toxic potency and carcinogenic action, Aflatoxin B1 (AFB1) is the most important AF (*Kew, 2013*). It causes multiple effects including mitochondrial permeability transition, DNA damage (*Shi et al., 2015*), oxidative stress (*Singh, Maurya & Trigun, 2015*), apoptosis (*Peng et al., 2016*), the defects in skeletal muscle development in different models

Corresponding author
Shao-Chen Sun, sunsc@njau.edu.cn

(*Gunduz & Oznurlu, 2014*). In porcine oocytes, AFB1 is shown to affect cell cycle and induce oxidative stress (*Liu et al., 2015*).

Deoxynivalenol that are produced mainly by fungi is a common contaminant of cereals including maize, wheat and barley (*Wu et al., 2011*). After feeding the domestic animals and poultry with DON-contaminated feed, it will exist in meat, milk, and eggs. In animals, it causes organ damage and hepatic lipid accumulation (*Pietsch et al., 2014*), emesis, anorexia, growth retardation, immunotoxicity as well as impaired reproduction and development competence (*Pestka, 2010*). At the cellular and molecular level, DON can induce apoptosis (*Li et al., 2014*), oxidative stress and genotoxicity (*Yang et al., 2014*). In addition, previous study showed that DON affected spindle morphology in porcine oocytes (*Han et al., 2016*).

HT-2 is the major metabolite of T-2 which is one of the type A trichothecene mycotoxins produced by different *Fusarium* species. HT-2 causes a myriad of effects including the inhibition of protein, DNA and RNA synthesis; oxidative stress (*Zhang et al., 2016*); reduced reproduction (*Zhu et al., 2016*) and embryo-fetal toxicity (*Wang et al., 2014*).

Several fungi including *Aspergillus ochraceus, A. carbonarius, A. niger*, and *Penicillium verrucosum* produce OTA. OTA is toxic to domestic animals, and its main target organ is kidney (*Grajewski et al., 2012*). OTA causes several effects like immunotoxicity (*Al-Anati & Petzinger, 2006*), hepatotoxicity, apoptosis, decrease of cell viability, impairment of mouse oocyte maturation and embryonic development (*Huang & Chan, 2016*).

Zearalenone produced by various *Fusarium* species is a contaminant of cereal crops and animal feed. Structurally, ZEA is similar to 17β-estradiol and it will compete with 17β-estradiol for binding to estrogen receptors, which leads to reproductive disorders (*Cortinovis et al., 2013*; *Minervini et al., 2001*, *2006*). The genotoxicity of ZEA has also been confirmed since it induces DNA fragmentation, apoptosis, and interruption of cell cycle progression (*Abid-Essefi et al., 2003*). Several studies showed that ZEA reduced porcine oocyte developmental competence (*Hou et al., 2014*; *Komsky-Elbaz, Saktsier & Roth, 2018*).

Although previous studies reported that several components of mycotoxins impaired mammalian oocyte quality. However, different effective concentrations were reported in the oocytes of different animal models. Although AFB1, DON, HT-2, OTA, and ZEA all have their metabolites in vivo, these mycotoxins are all detected in ovary, indicating that besides their metabolites, AFB1, DON, HT-2, OTA, and ZEA could directly affect ovary functions. In this study we aimed to compare the sensitivity of mouse and porcine oocytes to AFB1, DON, HT-2, OTA, and ZEA, which could provide the basic data for mycotoxin studies in the future. We adopted the polar body extrusion rate of mouse and porcine oocyte as the standard for the toxic effects of mycotoxins on oocyte maturation, and our results showed that except HT-2, porcine oocytes were generally more sensitive to AFB1, DON, OTA, and ZEA.

## MATERIALS AND METHODS

### Chemicals and regent

Aflatoxin B1, DON, HT-2, OTA, and ZEA were from J&K Chemical Ltd. (Shanghai, China). TCM-199 was from Gibco (Life Technologies, Carlsbad, CA, USA). TCM-199
which contained 75 µg/ml of penicillin, 50 µg/ml of streptomycin, 0.5 µg/ml of FSH, 0.5 µg/ml of LH, 10 ng/ml of the epidermal growth factor, and 0.57 mM cysteine was used for oocyte maturation. M2 and M16 culture medium were from Sigma-Aldrich (Merck; St. Louis, MO, USA).

## Oocytes collection and culture

We followed the guidelines of Animal Research Institute Committee of Nanjing Agricultural University to conduct the experiments (SYXK-Su-20170007). Germinal vesicle-intact oocytes of mice that obtained from the ovaries of three- to five-week old ICR mice were collected in M2 medium and cultured with M16 medium (Sigma Chemical Co., St. Louis, MO, USA) under the paraffin oil. These oocytes of mice were placed at 37 °C with 5% $CO_2$ for 12 h to observe the polar body extrusion.

Ovaries of Duroc pigs were purchased at a local slaughterhouse of Feng Yong Food Industry. After slaughter, ovaries were placed in a thermos bottle which contained 0.9% physiological saline and then delivered to our laboratory within 2 h. The temperature of the thermos bottle was close to 38 °C. Once the ovaries were delivered, they were washed with sterile saline. We aspirated follicular fluids from 3 to 6 mm antral follicles with a 10 ml disposable syringe and an 18 G needle. Cumulus oocyte complexes (COCs) with intact and compact cumulus were selected for maturation. These oocytes were placed at 38.5 °C with 5% $CO_2$ for 44 h to observe the polar body extrusion.

## Toxin treatment

AFB1, DON, HT-2, OTA, and ZEA were dissolved and stored at 50 mM in DMSO and then diluted into different concentrations with M199 or M16 maturation medium. The GV oocytes were then cultured with these mycotoxins to analyze the maturation rate by the polar body extrusion index. The same quantity of DMSO was added in the control group.

## Statistical analysis

Data are presented as means ($n = 3$). The concentration-response curves were made by GraphPad Prism 5. At least three biological replicates were used for each analysis. Each replicate was done by an independent experiment at the different time. Results are given as means ± SEM, and two groups were compared by student $t$-test. A $p$-value of <0.05 was considered significant.

# RESULTS

## Effects of AFB1 on mouse and porcine oocyte maturation

We first examined the effects of AFB1 on mouse and porcine oocytes. Mouse oocytes were cultured for 12 h with 10, 50, 100, and 150 µM AFB1. The average polar body extrusion rate of the control group was 79.94 ± 4.3% ($n = 142$) (Fig. 1A). Compared with control oocytes, when oocytes were cultured with 10 µM AFB1, rate of polar body extrusion showed no significant difference (72.82 ± 4.83%, $n = 110$, $p > 0.05$). Rates of

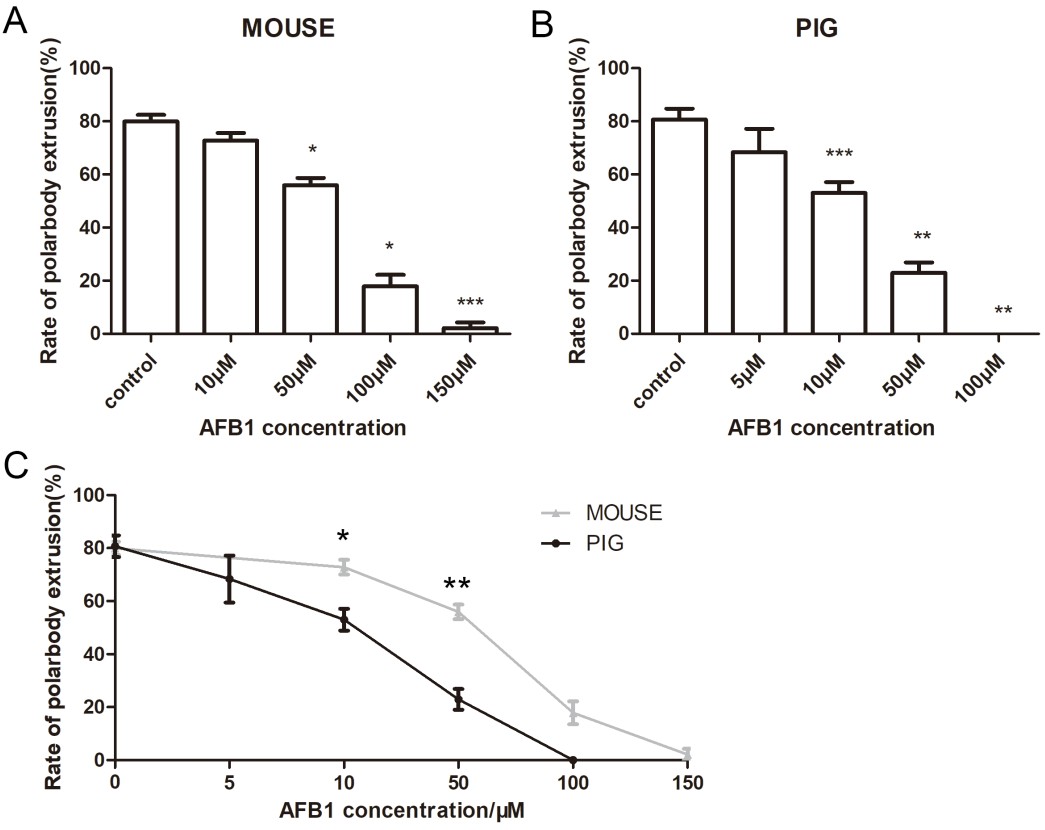

**Figure 1 The effects of AFB1 on the oocyte maturation.** (A) Rate of polar body extrusion after AFB1 treatment. Mouse oocytes were cultured with 0, 10, 50, 100, 150 μM AFB1 for 12 h. (B) Rate of polar body extrusion after AFB1 treatment. Porcine oocytes were cultured with 0, 5, 10, 50, 100 μM AFB1 for 44 h. Both polar body extrusion rates of mouse and porcine oocytes were reduced with the increasing concentration. (C) The statistical analysis for the effects of same concentration on mouse and porcine oocytes. The grey line represents the rate of mouse polar body extrusion. The black line represents the rate of porcine polar body extrusion. *, significance, $p < 0.05$; **, significance, $p < 0.01$; ***, significance, $p < 0.001$.               

matured oocytes were significantly decreased when the concentration of AFB1 were increased to 50 μM ($55.96 \pm 4.77\%$, $n = 157$, $p < 0.05$), 100 μM ($17.86 \pm 7.51\%$, $n = 154$, $p < 0.05$), 150 μM ($2.15 \pm 2.15\%$, $n = 146$, $p < 0.001$). We cultured the porcine oocytes for 44 h with 5, 10, 50, 100 μM AFB1. The polar body extrusion rate was $80.72 \pm 7.05\%$ ($n = 217$) in the control group of porcine oocytes, which was close to mouse oocytes. However, the rate of polar body extrusion in porcine oocytes was decreased with the increased AFB1 concentration. Porcine oocytes cultured with 5 μM AFB1 and the control oocytes showed no significant difference ($68.34 \pm 15.39\%$, $n = 195$ vs $80.72 \pm 7.05\%$ $n = 217$, $p > 0.05$). However, the rate of polar body extrusion in porcine oocytes was significantly decreased with 10 μM ($53.02 \pm 7.12\%$, $n = 144$, $p < 0.001$), 50 μM ($22.95 \pm 6.76\%$, $n = 192$, $p < 0.01$), 100 μM ($0 \pm 0\%$, $n = 147$, $p < 0.01$) AFB1 treatment (Fig. 1B). We also analyzed the mouse groups and porcine groups at the same concentrations. With the same concentration of AFB1 treatment, the rate of porcine oocyte polar body extrusion was lower than mouse oocytes. As shown in Fig. 1C, the rates

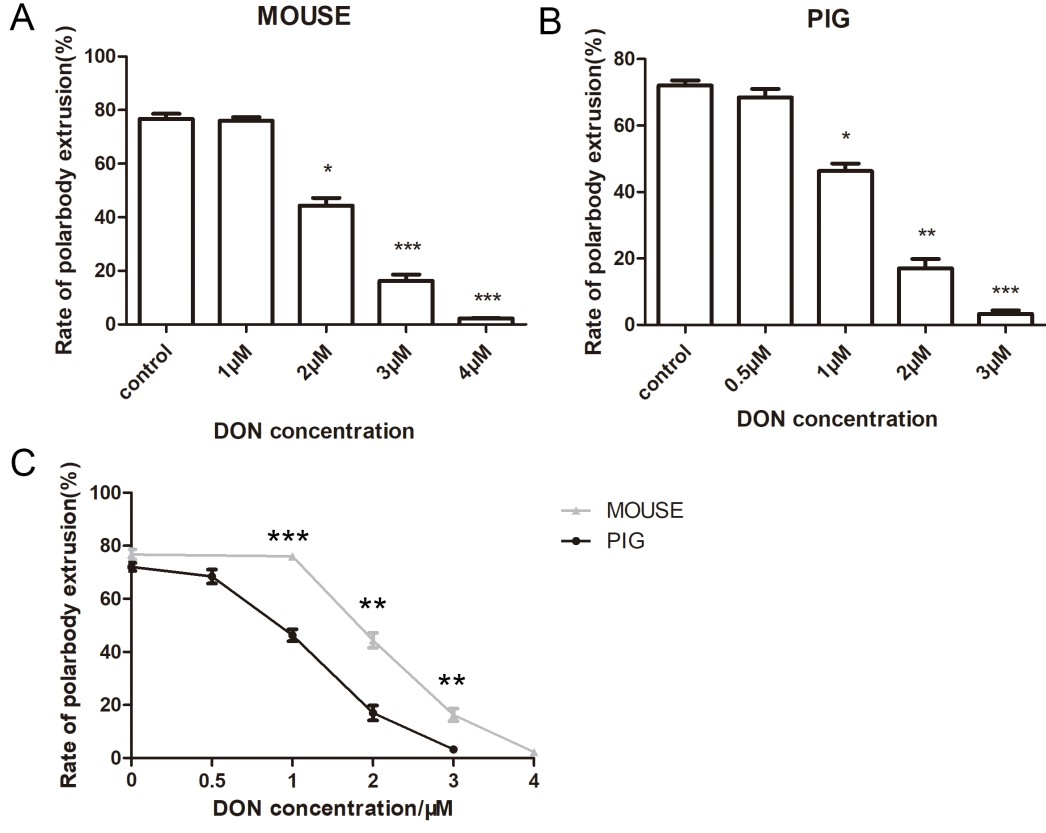

**Figure 2 The effects of DON on the oocyte maturation.** (A) Rate of polar body extrusion after DON treatment. For mouse oocyte culture, the DON concentration was 0, 1, 2, 3, 4 µM. (B) Rate of polar body extrusion after DON treatment. For porcine oocytes, DON concentration was 0, 0.5, 1, 2, 3 µM. The reduction of polar body extrusion was dose-dependent. (C) The statistical analysis for the effects of same concentration on mouse and porcine oocytes. The grey line represents the rate of mouse polar body extrusion. The black line represents the rate of porcine polar body extrusion. *, significance, $p < 0.05$; **, significance, $p < 0.01$; ***, significance, $p < 0.001$.              

of the mouse and porcine control groups showed no significant difference ($p > 0.05$), while there were significant differences when the concentrations were 10 µM ($p < 0.05$) and 50 µM ($p < 0.01$), indicating that compared with mouse oocytes, porcine oocytes were more sensitive to AFB1.

## Effects of DON on mouse and porcine oocyte maturation

We next examined the effects of DON on mouse and porcine oocytes. Mouse oocytes were cultured for 12 h with 1, 2, 3, and 4 µM DON. Our results showed that DON affected mouse oocyte maturation. The average polar body extrusion rate of the control group was 76.73 ± 3.24% ($n = 162$) (Fig. 2A), when mouse oocytes cultured with 1 µM AFB1, there was no significantly difference between the control group and 1 µM group (76.01 ± 2.35% $n = 139$, $p > 0.05$). While the rate of polar body extrusion in mouse oocytes was significantly decreased with 2 µM DON treatment (44.38 ± 4.87%, $n = 173$, $p < 0.05$), 3 µM DON treatment (16.30 ± 4.00%, $n = 199$, $p < 0.001$), 4 µM DON treatment (2.28 ± 0.69%, $n = 189$, $p < 0.001$), compared with the control group.

We cultured the porcine oocytes for 44 h with 0.5, 1, 2, 3 μM DON. The polar body extrusion rate was $72.05 \pm 2.6\%$ ($n = 195$) in the control group of porcine oocytes, which was close to mouse oocytes. At the concentration of 0.5 μM, the control group and 0.5 μM treatment group ($68.42 \pm 4.55\%$ $n = 159$, $p > 0.05$) showed no significant difference. However, the rate of polar body extrusion in porcine oocytes was significantly decreased with 1 μM DON treatment ($46.29 \pm 3.89\%$, $n = 176$, $p < 0.05$), 2 μM DON treatment ($17.02 \pm 4.87\%$, $n = 145$, $p < 0.01$), 3 μM DON treatment ($3.29 \pm 1.81\%$, $n = 132$, $p < 0.001$) (Fig. 2B). To compare the sensitivity of mouse and porcine, we also analyzed the rate of same concentrations. The rates of control groups were close to each other, while at the concentrations of 1 μM ($p < 0.001$), 2 μM ($p < 0.01$), 3 μM ($p < 0.01$), polar body extrusion rates of mouse oocytes and porcine oocytes showed significant difference (Fig. 2C), indicating that compared with mouse oocytes, porcine oocytes were more sensitive to DON.

## Effects of HT-2 on mouse and porcine oocyte maturation

We next examined the effects of HT-2 on mouse and porcine oocytes. The mouse oocytes were cultured for 12 h with 10, 20, 30, and 40 nM HT-2. The average polar body extrusion rate of the control group was $73.08 \pm 1.67\%$ ($n = 153$) (Fig. 3A), while the rate of polar body extrusion in mouse oocytes was significantly decreased with 10 nM HT-2 treatment ($43.33 \pm 4.93\%$, $n = 176$, $p < 0.05$), 20 nM HT-2 treatment ($33.05 \pm 2.18\%$, $n = 163$, $p < 0.001$), 30 nM HT-2 treatment ($7.08 \pm 0.89\%$, $n = 150$, $p < 0.001$), 40 nM HT-2 treatment ($1.05 \pm 0.10\%$, $n = 169$, $p < 0.001$). The porcine oocytes were cultured for 44 h with 10, 50, 100, 150, 200, and 400 nM HT-2. Similar with the mouse oocytes, our results showed that HT-2 affected porcine oocyte maturation. The polar body extrusion rate was $78.19 \pm 2.03\%$ ($n = 171$) in the control group of porcine oocytes, which was close to mouse oocytes. However, except the 10 nM HT-2 treatment ($68.67 \pm 2.99\%$, $n = 163$, $p > 0.05$), the rate of polar body extrusion in porcine oocytes was significantly decreased with 50 nM HT-2 treatment ($47.4 \pm 3.36\%$, $n = 156$, $p < 0.01$), 100 nM HT-2 treatment ($25.50 \pm 5.14\%$, $n = 164$, $p < 0.01$), 150 nM HT-2 treatment ($21.22 \pm 4.07\%$, $n = 188$, $p < 0.01$), 200 nM HT-2 treatment ($18.02 \pm 6.69\%$, $n = 175$, $p < 0.01$), 400 nM HT-2 treatment ($8.22 \pm 0.78\%$, $n = 162$, $p < 0.001$) (Fig. 3B). The control groups in mouse and porcine have similar values, however, at the concentration of 10 μM ($p < 0.01$) rates of mouse and porcine were significantly different (Fig. 3C). Our results showed that with the same concentration of HT-2 treatment the rate of mouse oocyte polar body extrusion was lower than porcine oocytes, indicating that compared with porcine oocytes mouse oocytes were more sensitive to HT-2.

## Effects of OTA on mouse and porcine oocyte maturation

We next examined the effects of OTA on mouse and porcine oocytes. Mouse oocytes were cultured for 12 h with 200, 300, 400, and 600 μM OTA. The average polar body extrusion rate of the control group was $80.23 \pm 3.87\%$ ($n = 169$) (Fig. 4A), compared with the control group, the concentration of 200 μM OTA showed no significant difference ($73.77 \pm 2.24\%$ $n = 143$, $p > 0.05$), while the rate of polar body extrusion in mouse

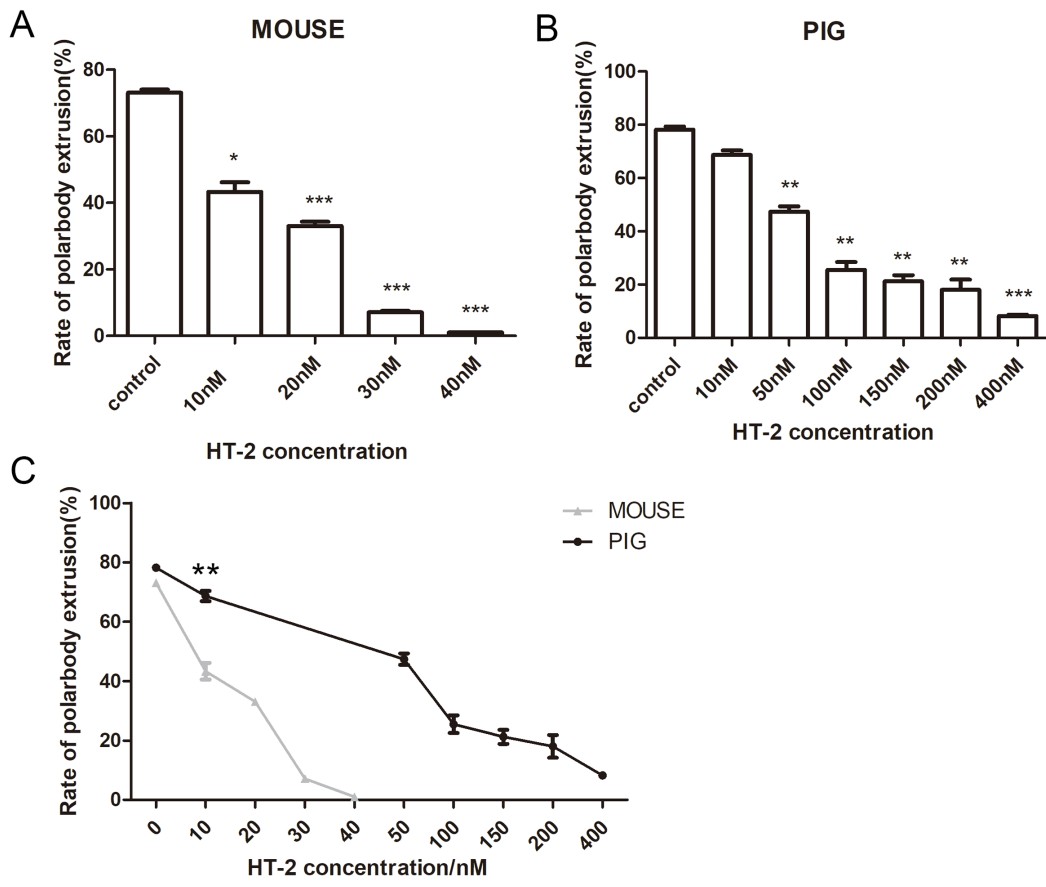

**Figure 3 The effects of HT-2 on the oocyte maturation.** (A) Rate of polar body extrusion after HT-2 treatment. Mouse oocytes were cultured with HT-2 at the concentration of 0, 10, 20, 30, 40 nM. (B) Rate of polar body extrusion after HT-2 treatment. For porcine oocytes culture, HT-2 was 0, 10, 50, 100, 150, 200, 400 nM. The rate of polar body extrusion decreased when the concentration of HT-2 increased. (C) The statistical analysis for the effects of same concentration on mouse and porcine oocytes. The grey line represents the rate of mouse polar body extrusion. The black line represents the rate of porcine polar body extrusion. *, significance, $p < 0.05$; **, significance, $p < 0.01$; ***, significance, $p < 0.001$.

oocytes was significantly decreased with 300 μM OTA treatment ($49.86 \pm 4.29\%$, $n = 190$, $p < 0.01$), 400 μM OTA treatment ($31.23 \pm 3.64\%$, $n = 145$, $p < 0.01$), 600 μM OTA treatment ($2.22 \pm 2.22\%$, $n = 178$, $p < 0.01$). The porcine oocytes were cultured for 44 h with 1, 5, 10, 30, and 100 μM OTA. Similar with the mouse oocytes, our results showed that OTA affected porcine oocyte maturation. The control group and 1 μM OTA group showed no significant difference ($80.34 \pm 7.95\%$ $n = 164$ vs $65.46 \pm 4.09\%$ $n = 178$, $p > 0.05$). However, the rate of polar body extrusion in porcine oocytes was significantly decreased with 5 μM OTA treatment ($53.83 \pm 0.34\%$, $n = 154$, $p < 0.05$), 10 μM OTA treatment ($22.26 \pm 3.14\%$, $n = 183$, $p < 0.05$), 30 μM OTA treatment ($22.19 \pm 4.87\%$, $n = 140$, $p < 0.01$), 100 μM OTA treatment ($4.24 \pm 3.93\%$, $n = 144$, $p < 0.01$) (Fig. 4B). With the same concentration of OTA treatment, the rate of porcine oocyte polar body extrusion was lower than mouse oocytes (Fig. 4C), indicating that compared with mouse oocytes, porcine oocytes were more sensitive to OTA.
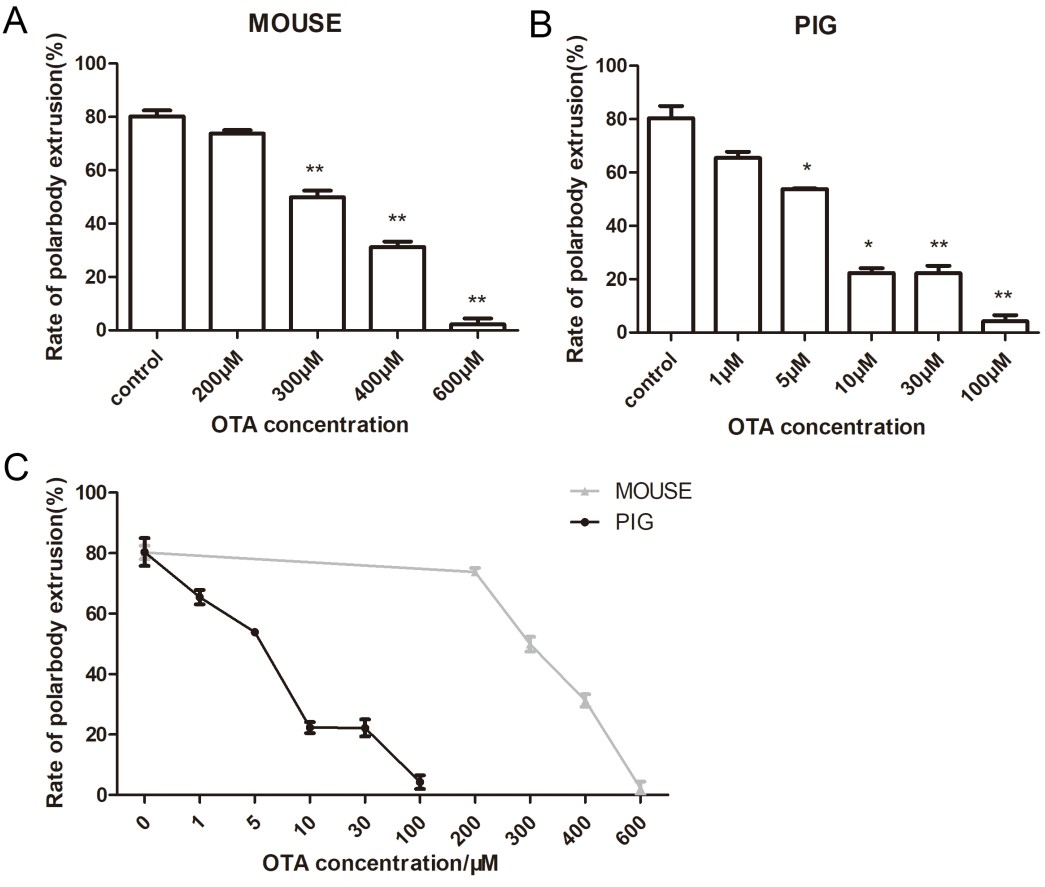

**Figure 4 The effects of OTA on the oocyte maturation.** (A) Rate of polar body extrusion after OTA treatment. For mouse oocyte culture, OTA concentration was 0, 200, 300, 400, 600 μM. (B) Rate of polar body extrusion after OTA treatment. For porcine oocyte culture, OTA concentration was 0, 1, 5, 10, 30, 100 μM. *, significance, $p < 0.05$; **, significance, $p < 0.01$. (C) The comparison for the effects of different concentrations on mouse and porcine oocytes. The grey line represents the rate of mouse polar body extrusion. The black line represents the rate of porcine polar body extrusion.

## Effects of ZEA on mouse and porcine oocyte maturation

The last we examined was the effects of ZEA on mouse and porcine oocytes. Mouse oocytes were cultured for 12 h with 10, 50, 100, and 200 μM ZEA. The average polar body extrusion rate of the control group was 81.29 ± 6.06% ($n = 155$) (Fig. 5A), when the concentration was 10 μM, the average rate of MII oocytes was 74.52 ± 4.92% ($n = 154$) which showed no significant difference with the control groups ($p > 0.05$). While the rate of polar body extrusion in mouse oocytes was significantly decreased with 50 μM ZEA treatment (54.35 ± 3.9%, $n = 128$, $p < 0.05$), 100 μM ZEA treatment (26.23 ± 8.00%, $n = 150$, $p < 0.05$), 200 μM ZEA treatment (0.00 ± 0.00%, $n = 132$, $p < 0.01$). For porcine oocytes, the average polar body extrusion rate of the control groups was 77.85 ± 9.51% ($n = 175$), while the 5 μM ZEA groups (60.45 ± 1.65%, $n = 199$, $p > 0.05$) showed no significant differences with the control groups. However, the rate of polar body extrusion in porcine oocytes was significantly decreased with the 10 μM groups

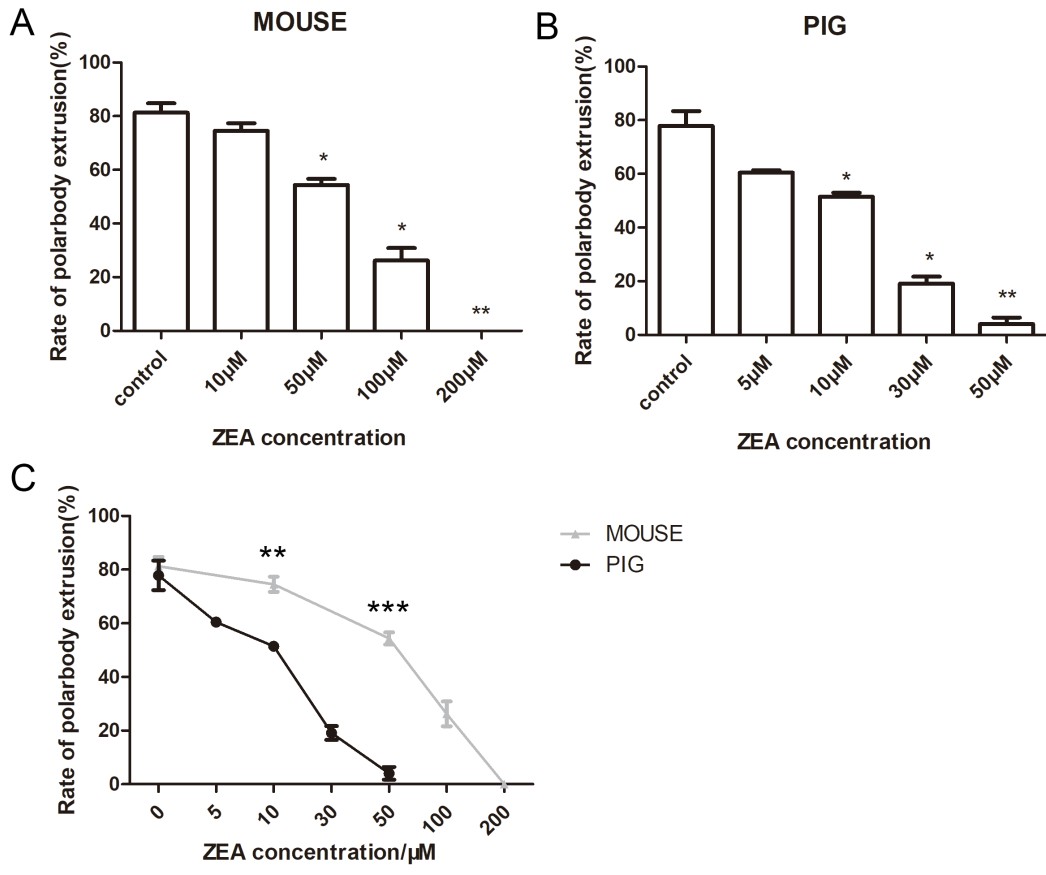

**Figure 5 The effects of ZEA on the oocyte maturation.** (A) Rate of polar body extrusion after ZEA treatment. Mouse oocytes were cultured with 0, 10, 50, 100, 200 μM ZEA. (B) Rate of polar body extrusion after ZEA treatment. Porcine oocytes were cultured with 0, 5, 10, 30, 50 μM ZEA. Both mouse and porcine polar body extrusion rate were reducing when ZEA concentration was increasing. (C) The statistical analysis for the effects of same concentration on mouse and porcine oocytes. The grey line represents the rate of mouse polar body extrusion. The black line represents the rate of porcine polar body extrusion. *, significance, $p < 0.05$; **, significance, $p < 0.01$; ***, significance, $p < 0.001$.

($51.42 \pm 2.73\%$, $n = 190$, $p < 0.05$), 30 μM ZEA treatment ($19.10 \pm 4.49\%$, $n = 147$, $p < 0.05$) and 50 μM ZEA treatment ($4.08 \pm 4.08\%$, $n = 148$, $p < 0.01$) (Fig. 5B). Our results showed that with the same concentration of ZEA treatment, the rate of porcine oocyte polar body extrusion was lower than mouse oocytes. Control groups of mouse and porcine oocytes showed no significant difference ($p > 0.05$). However, there were significant difference between mouse oocytes and porcine oocytes at 10 μM ($p < 0.01$) and 50 μM ($p < 0.001$) (Fig. 5C), indicating that compared with mouse oocytes, porcine oocytes were more sensitive to ZEA.

## DISCUSSION

In the present study we used the polar body extrusion as the index for oocyte maturation to compare the sensitivity of mouse and porcine oocytes to AFB1, DON, HT-2, OTA, and ZEA. Our results provided the basic database for the mycotoxin on oocyte studies.

Mycotoxins are shown to affect human and animal health from multiple aspects, such as immune system, micro-organisms. And recent years, the toxicity of mycotoxins on reproductive system especially on oocytes and sperms were reported. Our previous work found that when 50 µM AFB1 affected COCs growth, especially the polar body extrusion was significantly reduced in porcine oocytes (*Liu et al., 2015*). However, 10 µM AFB1 significantly increased the proportion of sperm with fragmented DNA in mice (*Komsky-Elbaz, Saktsier & Roth, 2018*). This indicated that even in the reproductive system, the sensitivity of different cell types or animal models to the mycotoxins is different. Our results showed that 10 µM AFB1 affected porcine oocyte maturation instead of 50 µM AFB1 in mouse oocytes.

A total of 2 µM DON was shown to affect the formation of the meiotic spindle in mouse oocytes (*Lan et al., 2018*). While a recent study showed that 10 µM DON affected the morphology of pig ovaries with an ex vivo approach (*Gerez, Desto & Bracarense, 2017*). Our recent study also showed that 3 µM DON exposure altered autophagy/apoptosis and epigenetic modifications in porcine oocytes (*Han et al., 2016*). In the present study our results showed that 1 µM DON already affected porcine oocyte maturation instead of 2 µM DON in mouse oocytes, which showed similar sensitivity pattern to AFB1. However, HT-2 had different sensitivity pattern compared with AFB1 and DON. HT-2 toxin was shown to affect cytoskeletal dynamics, apoptosis/autophagy, oxidative stress, and epigenetic modifications in mouse oocytes (*Zhu et al., 2016*). Our results showed that 10 nM HT-2 affected mouse oocyte maturation while the similar results only occurred at 50 nM HT-2 exposure for porcine oocytes. Further study is still needed to explore the toxic effects of HT-2 toxin in different reproductive cell types like cumulus cells and sperm.

A recent study indicated that OTA significantly impaired oocyte maturation, in vitro fertilization (IVF) rates and inhibited embryonic development in vitro, because OTA could induce caspase-dependent apoptosis with in vivo model (*Huang & Chan, 2016*). 1–10 µM OTA in the drinking water was adopted in this study. Our results showed that 5 µM OTA affected porcine oocyte maturation instead of 300 µM OTA in mouse oocytes. The big difference for the OTA in mouse oocyte between in vivo and in vitro model needs more study to explain. For ZEA, at the concentration of 30 µM, ZEA was shown to affect porcine oocyte maturation and embryonic development through oxidative stress, autophagy and early apoptosis (*Komsky-Elbaz, Saktsier & Roth, 2018*); and for mouse oocytes, it affected oocyte quality by altering the epigenetic modification levels (*Zhu et al., 2014*). Our results showed that 10 µM ZEA affected porcine oocyte maturation instead of 50 µM ZEA in mouse oocytes, which showed similar concentration pattern with AFB1 and DON.

## CONCLUSION

In all, our results showed that these five mycotoxins all affected mouse and porcine oocyte quality, however, different sensitivity patterns between mouse and porcine oocytes were found. Generally porcine oocytes were more sensitive to AFB1, DON, OTA, ZEA

compared with mouse oocytes except HT-2. Our results provided a basic database for the further studies on mammalian oocytes.

### Funding

This work was supported by the National Basic Research Program of China (2014CB138503); the National Natural Science Foundation of China (31622055, 31571547); the Fundamental Research Funds for the Central Universities (KYTZ201602, KJYQ201701), China. The funders had no role in study design, data collection and analysis, decision to publish, or preparation of the manuscript.

### Grant Disclosures

The following grant information was disclosed by the authors:
National Basic Research Program of China: 2014CB138503.
National Natural Science Foundation of China: 31622055, 31571547.
Fundamental Research Funds for the Central Universities: KYTZ201602, KJYQ201701.

### Competing Interests

Shao-Chen Sun is an Academic Editor for PeerJ. Yong-Teng Su and Yu-Rong Sun are employed by Jiangsu Aomai Bio-tech Company.

### Author Contributions

- Yujie Lu conceived and designed the experiments, performed the experiments, analyzed the data, prepared figures and/or tables, authored or reviewed drafts of the paper, approved the final draft.
- Yue Zhang conceived and designed the experiments, performed the experiments, analyzed the data, prepared figures and/or tables, approved the final draft.
- Jia-Qian Liu contributed reagents/materials/analysis tools, approved the final draft.
- Peng Zou contributed reagents/materials/analysis tools, approved the final draft.
- Lu Jia contributed reagents/materials/analysis tools, approved the final draft.
- Yong-Teng Su contributed reagents/materials/analysis tools, approved the final draft.
- Yu-Rong Sun contributed reagents/materials/analysis tools, approved the final draft.
- Shao-Chen Sun conceived and designed the experiments, analyzed the data, prepared figures and/or tables, authored or reviewed drafts of the paper, approved the final draft.

### Animal Ethics

The following information was supplied relating to ethical approvals (i.e., approving body and any reference numbers):

The Animal Research Institute Committee of Nanjing Agricultural University provided full approval for this study (SYXK-Su-20170007).

## Data Availability

The raw data are provided in a Supplemental File.

## Supplemental Information

Supplemental information for this article can be found online at http://dx.doi.org/10.7717/peerj.5111#supplemental-information.

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
