# Peer review of "Comparison of the toxic effects of different mycotoxins on porcine and mouse oocyte meiosis"

_PeerJ, doi:10.7717/peerj.5111_

## Round 0.1 · original submission · Minor Revisions

Your manuscript was well written although some questions need to be modified. Please revise your manuscript according to the reviewers' suggestion.

·

Basic reporting

no comment

Experimental design

no comment

Validity of the findings

no comment

Additional comments

This is a very interesting study that is additive to our knowledge of different mycotoxins impact on porcine and mouse oocyte meiosis as well as has provided basic data for the further studies on mammalian oocyte research. The authors isolated oocyte from antral follicles of pigs and mice and exposed those in culture to different concentrations of different mycotoxins. They determined the different sensitivity pattern between mouse and porcine oocytes. In manuscript, the grammar and expression of English would be better to get a major revision to make readers' easy understanding. There are also minor comments to address:
1. Since dosing is done in vitro, the authors need to provide a justification for the doses selected.
2. Were all of the porcine and mouse oocyte collected and cultured at the same time? How many oocytes did they collected used for statistic?
3. 206, 209,223,229=10 μM, not 10μM. Please recheck and correct the related contents.

Reviewer 2 ·

Basic reporting

'no comment'

Experimental design

'no comment'

Validity of the findings

'no comment'

Additional comments

The manuscript entitled “Comparison of the toxic effects of different mycotoxins on porcine and mouse oocyte meiosis” submitted by Lu et al. describes a study about the effects of mycotoxins (Aflatoxin B1, Deoxynivalenol, HT-2, ochratoxin A, Zearalenone) on porcine and mouse oocyte maturation. The results showed that all of these mycotoxins compromised the porcine and mouse oocyte maturation. The authors conclude that porcine oocytes were more sensitive to Aflatoxin B1, Deoxynivalenol, ochratoxin A and Zearalenone than mouse oocytes except HT-2 toxin. The objective of the study is clear and this manuscript contains sufficient new information, although information concerning the method, and interpretation of results shows some weakness. This reviewer only has some specific remarks for the improvement of the paper before its publication in Peer J.

Major comments
1. The authors subject experimental data of oocyte maturation rate to two-tailed unpaired student’s t-test. However, experimental data shown in Figures 1 to 5 have to analyze using one-way ANOVA and the post hoc test. In addition, please add significant markers in Figures.
2. From Figure 1 to 5, the table repeats the same data as the concentration-response curves. Please remove this table, or you can show this table in supplement data.
3. In each part of results, the author descripted “the rate … of oocyte polar body extrusion was lower than …”, but no significant analysis was carried out.
4. Titles of figure 2 to 5 were not matched with the figure legend and results. Please check and rewrite.

Minor comments
Line 45-46: If mycotoxins are “important” in agriculture? Please, revise this lines.
Line 49: Please change “exists” to “exist”.
Line 51-52: What is “mitochondrial permeability transition DNA damage”? Please, revise this lines.
Line 54: Please change “induced” to “induce”.
Line 83: Please change “two” to “twice”.
Line 106: Please delete “purchased”.
Line 109: If “2 to 8mm antral follicles” was used in your study? Please make sure of this. Because 3 to 6 mm antral follicles were usually used.
Lines 221, …: In vivo, in vitro and ex vivo should be written in italic.
Line 211: Please delete the comma.
In Figure 3 and 4, there are radix point after the number of HT-2 and OTA concentration, please revise these figure.

Reviewer 3 ·

Basic reporting

no comment

Experimental design

no comment

Validity of the findings

no comment

Additional comments

In the manuscript entitled “Comparison of the toxic effects of different mycotoxins on porcine and mouse oocyte meiosis”, the authors performed a systematic investigation of different concentrations of AFB1, DON, HT-2, OTA and ZEA in oocyte maturation, using both mouse and pig as the in vitro model, focusing on the first polar body extrusion rate.

Overall, the manuscript is well written and technically sound, the results are clearly demonstrated as well. It’s also interesting and significant as this is the first study that systematically compared the sensitivity of mouse and porcine oocytes to AFB1, DON, HT-2, OTA and ZEA for mycotoxin research. However, there are several minor issues based on the current manuscript and comments are outlined below.


1. In the body, the drug or chemical needs go through several layers of cells to work on oocytes, for example, theca cells, granulosa cells and cumulus cells (when follicle is grown up), and maybe even stromal cells. Authors are suggested to discuss, whether these 5 mycotoxins, AFB1, DON, HT-2, OTA and ZEA, are the final metabolites in vivo?
2. Please explain why use different maturation medium. It seems mouse oocytes can also be cultured/matured in TCM199?
3. Please add more details. For example, “from prepubertal gilts purchased at a local slaughterhouse”, name and address; “in a thermos bottle”, temperature?
4. It is suggested to put all 5 mycotoxins as Keywords (may use “X and X” to decrease the number).
5. “The most agriculturally important mycotoxins known today are aflatoxins (AF), Deoxynivalenol (DON), T-2, ochratoxin A (OTA), Zearalenone (ZEA)” needs reference(s).
6. Authors are suggested to carefully go through the manuscript and correct minor grammatical errors, for instance, “effects including reproductive system on animals and humans.”; “previous studies reported that several components of mycotoxins impaired mammalian oocyte quality. However, different effective concentrations were reported in the oocytes of different animal models”.

Reviewer 4 ·

Basic reporting

no comment

Experimental design

no comment

Validity of the findings

no comment

Additional comments

There are a few comments for the improvement:
1. The English language should be improved.
2. In “Toxin treatment” of Materials and Methods, the details of treatment were missing.
3. The authors should mention the standard for the “……significantly affected……” of these mycotoxins in Results or Discussion.
4. In “Effects of AFB1 on mouse and porcine oocyte maturation” of Results, the authors described the effects of 100 μM AFB1 treatment in mouse oocytes and 50 μM AFB1 treatment in porcine oocytes, but they mention “Our results showed that 10μM AFB1 affected porcine oocyte maturation instead of 50μM AFB1 in mouse oocytes.” in Discussion. Please explain.
5. Discussion needs to be improved.

---

## Round 0.2 · accepted · Accept

I can confirm that the manuscript was revised according to the reviewers' suggestion.

#